# Higher Accuracy of Lung Ultrasound over Chest X-ray for Early Diagnosis of COVID-19 Pneumonia

**DOI:** 10.3390/ijerph18073481

**Published:** 2021-03-27

**Authors:** Javier Martínez Redondo, Carles Comas Rodríguez, Jesús Pujol Salud, Montserrat Crespo Pons, Cristina García Serrano, Marta Ortega Bravo, Jose María Palacín Peruga

**Affiliations:** 1Balaguer Primary Care Center, Institut Català de la Salut (ICS), 25600 Lleida, Spain; jmartinez.lleida.ics@gencat.cat (J.M.R.); jpujol.lleida.ics@gencat.cat (J.P.S.); mcrespo.lleida.ics@gencat.cat (M.C.P.); cgarcia.lleida.ics@gencat.cat (C.G.S.); 2Department of Mathematics, Campus Cappont, University of Lleida, 25001 Lleida, Spain; carles.comas@udl.cat; 3Biomedical Research Institute (IRB Lleida), Universitat de Lleida (UdL), 25198 Lleida, Spain; 4Research Group in Therapies in Primary Care (GRETAPS), 25007 Lleida, Spain; 5Research Support Unit Lleida, Fundació Institut Universitari per a la recerca a l’Atenció Primària de Salut Jordi Gol i Gurina (IDIAPJGol), 25007 Lleida, Spain; 6Onze de Setembre Primary Care Center, Institut Català de la Salut (ICS), Passeig Onze de Setembre, 10, 25005 Lleida, Spain

**Keywords:** general practice, lung ultrasonography, thoracic radiography, COVID-19

## Abstract

Background: The COVID-19 pandemic rapidly strained healthcare systems worldwide. The reference standard for diagnosis is a positive reverse transcription polymerase chain reaction (RT-PCR) test, but results are not immediate and sensibility is variable. Aim: To evaluate the diagnostic accuracy of lung ultrasound compared to chest X-ray for COVID-19 pneumonia. Design and Setting: A retrospective analysis of symptomatic patients admitted into one primary care centre in Spain between March and September 2020. Method: Patients’ chest X-rays and lung ultrasounds were categorized as normal or pathologic. RT-PCR confirmed COVID-19 infection. Pathologic lung ultrasound images were further categorized as showing either local or diffuse interstitial disease. McNemar and Fisher tests were used to compare diagnostic accuracy. Results: Most of the 212 patients presented fever at admission, either as a standalone symptom (37.74% of patients) or together with others (72.17% of patients). The positive predictive value of the lung ultrasound was 90% for the diffuse interstitial pattern and 46.92% for local pattern. The lung ultrasound had a significantly higher sensitivity (82.75%) (*p* < 0.001), but lower specificity (71%) than the chest X-ray (54.02% and 86%, respectively) (*p* = 0.008) for identifying interstitial lung disease. Moreover, sensitivity of the lung ultrasound for severe interstitial disease was 100%, and was significantly higher than the chest X-ray (58.33%) (*p* = 0.002). Conclusion: The lung ultrasound is more accurate than the chest X-ray for identifying patients with COVID-19 pneumonia and it is especially useful for those presenting diffuse interstitial disease.

## 1. Introduction

The recent COVID-19 virus arose in China in December 2019 and rapidly strained healthcare systems worldwide, having a severe economic and psychological impact [1]. On 11 March 2020, the World Health Organization declared COVID-19 a pandemic [2], which began noticeably impacting Spain, with 2128 cases and 47 deaths [3]. By 31 March 2020, these had increased to 94,417 cases and 8189 deaths [4].

During the first several months of the outbreak, diagnosis of the disease was challenging, due to the varying severity of the disease, the heterogeneity in the symptoms presented, and the lack of radiologic findings in many cases [5,6]. Quickly diagnosing COVID-19 became of the utmost importance to isolate and treat the affected patients, and to trace and test their close contacts. The high number of suspected cases and the increased occupancy of intensive care units severely impacted healthcare centers [7,8,9]. 

The reference standard for the COVID-19 diagnosis is the RT-PCR test, but it has variable sensibility, with up to 54% of infected patients having an initial negative result [10,11] and it is not able to assess the severity of the infection. These findings highlight the need for repeated RT-PCR testing, but also for identifying tools able to enhance or complement the diagnosis may be at specific stages of the disease. A known consequence of COVID-19 infection is the development of interstitial lung disease [12] and the Spanish Society of Medical Radiology published in March 2020 a guide recommending imaging methods for its detection [13]. This guide stated that these techniques, such as computed tomography, chest X-ray and lung ultrasound could help and complement the RT-PCR’s diagnosis. Computed tomography is the most sensible and specific, but it is not available in primary care; X-rays have been useful for triaging patients during the pandemic, but they are difficult to evaluate or unspecific [14,15,16] and, although quick, they expose patients to radiation [17]. A lung ultrasound was regarded as useful due to its high specificity and sensitivity.

A lung ultrasound is rapidly available, can be used at bedside and is highly versatile [17]. Although an ultrasound is not a tool traditionally used by general practitioners (GPs), its use has increased worldwide recently [18]. GPs have used ultrasound for a variety of indications, including the detection of diffuse interstitial syndrome in the lungs with high sensitivity and specificity [18]. Volpicelli et al. made recommendations in 2012 mentioning lung ultrasound should be considered for diagnosis of interstitial lung syndrome [19]; Volpicelli et al. also recently stated the interest of lung ultrasound in the COVID-19 pandemic [20]. The 2012 recommendations stated lung ultrasound was superior to the conventional chest X-ray for ruling in and out interstitial syndrome. Despite the existing learning curve for carrying out and interpreting an ultrasound, studies have shown there is high inter-rater agreement in ultrasound interpretation between GPs and specialists [21]. This was also found in Spain after GPs underwent specific training [22]. Several studies have reported lung ultrasound is a valuable diagnostic tool for COVID-19 pneumonia; however these were carried out in the hospital setting, most commonly in emergency room or intensive care units [23,24,25,26,27,28].

The aim of this study is to evaluate the usefulness of lung ultrasound compared to chest X-ray in early detection of interstitial lung disease caused by COVID-19 infection in primary care. By comparing their sensitivity, specificity, positive predictive value (PPV) and negative predictive value (NPV) analyzed by the same GP, it was assessed whether lung ultrasound may dismiss COVID-19 infection more accurately than the chest X-ray.

## 2. Materials and Methods 

### 2.1. Study Design and Participants

This single observational descriptive study was carried out at Balaguer primary care centre (Balaguer, Spain). A consecutive sample of patients admitted into emergency room of this primary care centre between 16 March 2020 and 30 September 2020 was retrospectively analyzed. Inclusion criteria were: (a) all ages (after 29 April 2020, ages 0–9 or over 18); (b) availability of lung ultrasound imaging; (c) suspected COVID-19 pneumonia prior to RT-PCR results, defined as presence of at least one of the compatible symptoms (cough, dyspnoea, fever as body temperature > 37 °C, oxygen saturation < 95%) and (d) consent to participate in the study. There were no exclusion criteria. Age criterion was modified in April to avoid causing discomfort to patients, since there were disagreements with patients in the 10–18 age group and their families.

We collected patients’ data from electronic medical records: demographic variables (age and gender), clinical variables (progress and symptoms), analytical variables (laboratory blood work), RT-PCR test from the nasopharyngeal swab, chest X-ray analysis (normal/abnormal) and lung ultrasound analysis.

### 2.2. Imaging Protocol and Interpretation

The lung ultrasound was conducted by scanning 4 zones (craniocaudal lines) per lung. A total of 8 zones were scanned in the posterior position (per lung: parallel and 5 cm from thoracic spine, medial border of scapula, posterior axillary line and medial axillary line). A convex probe CH5-2 (linear probe VF10-5 for children or very small-framed patients) with a bandwidth of 2–5 MHz (5–10 MHz for linear probe) was used on a Siemens Acuson X150 (Seoul, Korea) system. The exam was performed using the B mode with a general setting (abdominal) focused at the pleural line level (1–3 cm under, in case of obese patients) and a depth set to 6–12 cm.

The chest X-ray and lung ultrasound of each patient were interpreted by the same GP. Chest X-ray images were categorized as normal or abnormal (i.e., pathologic). Ultrasound images were categorized based on the overall interpretation and B-line pattern: Normal (no B-lines, or B-lines that cannot be categorized as the local or diffuse pattern), diffuse (two or more regions with more than 3 B-lines, bilaterally) or local (bat sign, two or more regions with more than 3 B-lines on a longitudinal axis between 2 ribs and not fulfilling the diffuse definition). Severe interstitial lung disease was defined by clinical (dyspnoea and/or oxygen saturation < 95%) and image findings (pathologic X-ray and/or diffuse or local pattern observed in ultrasound images) that led to hospitalization.

### 2.3. Ethical Approval

This study was approved by the Ethical Committee of Institut d’Investigació en Atenció Primària Jordi Gol i Gurina (Barcelona, Spain) (registration number p20/138).

### 2.4. Statistical Analysis

All 212 patients underwent lung ultrasound; of these, 187 also had a chest X-ray. A study with paired data was performed in these 187 patients with ultrasound and chest radiography. Diagnostic accuracy of the COVID-19 lung disease was compared between the two diagnostic methods using a nonparametric McNemar test. Fisher’s test was used to assess independence between lung ultrasound, chest radiography and RT-PCR test. Hospitalization versus non-hospitalization of patients was used in a Chi-square test as gold standard to compare the two imaging methods with respect to the detection of severe interstitial disease. Moreover a Chi-square test was also used for the bivariate analysis between the different clinical variables and results of RT-PCR. Additionally, a multivariate Chi-square test was used to determine relationships between the different clinical variables, imaging tests and RT-PCR’s results. Finally for comparison between the different ultrasound patterns with RT-PCR’s results. *p* < 0.05 was considered statistically significant in all tests. Statistical analyses were performed using the R statistical package, version 4.0.2. (R Core Team, 2020).

## 3. Results

### 3.1. Patient Demographics and Analysis of Symptoms

A total of 212 patients were included in this study (Figure 1) with a mean age of 49.62; 58.49% (124/212) of them were male (Table 1). The most common symptom overall was fever in 72.17% (153/212) of patients; fever was also the most common standalone symptom in 37.74% (80/212) of patients (Table 1). Only 3.77% (8/212) of patients presented at admission all four symptoms associated with COVID-19 considered in this study (cough, dyspnoea, fever as temperature > 37 °C and oxygen saturation < 95%).

Chi-squared analysis revealed that positive RT-PCR results were significantly associated with fever (χ^2^ = 18.74, *p* < 0.001), oxygen saturation < 95% (χ^2^ = 11.53, *p* < 0.001) and fever and cough presented together (χ^2^ = 19.90, *p* < 0.001). When evaluating the contingency tables, it was found that there was a trend for patients with a diffuse ultrasound pattern to present positive RT-PCR results, regardless of fever (Appendix A). Patients with fever also tended to be RT-PCR-positive when presenting a diffuse pattern in the lung ultrasound. In the absence of a diffuse pattern, patients with fever observed a tendency towards negative RT-PCR results. These results imply that diagnostic imaging findings have a higher correlation with RT-PCR than the most common symptom, fever.

### 3.2. Lung Ultrasound Achieved Higher PPV for Diffuse Interstitial Lung Disease Than for the Local Pattern

The PPV of lung ultrasound was 90% for diffuse interstitial pattern and 46.92% for the local pattern (Table 2). Additionally, Chi-squared analysis revealed no significant association between the COVID-19 infection confirmed by RT-PCR and local interstitial disease (χ^2^ = 0.0126, *p* = 0.9105); but there was a significant association with diffuse interstitial disease (χ^2^ = 60,1938, *p* < 0.001). These data imply that a lung ultrasound has a good PPV for diffuse interstitial lung disease caused by COVID-19 infection.

### 3.3. Lung Ultrasound Achieved Higher PPV for COVID-19 Diagnosis Than the Chest X-ray

Chi-squared analysis revealed that abnormal lung ultrasound findings were significantly associated with COVID-19 PCR results (χ^2^ = 47.58, *p* < 0.001); abnormal chest X-ray findings were also significantly associated with COVID-19 PCR results (χ^2^ = 33.91, *p* < 0.001). Abnormal findings with lung ultrasound and/or chest X-ray and RT-PCR results are shown in Table 3.

When analyzing paired samples, the lung ultrasound had a significantly higher sensitivity (82.75%) than the chest X-ray (54.02%) (*p* < 0.001) for interstitial disease and the chest X-ray had a significantly higher specificity (86.00% vs. 71.00%) (*p* = 0.008) (Table 4).

### 3.4. Lung Ultrasound Achieved 100% Sensitivity for Severe Interstitial Disease

It was next evaluated the diagnostic accuracy of both methods for the severe interstitial disease (i.e., hospitalized patients). Out of 187 patients who had received both ultrasound and chest X-ray, 24 (12.83%) presented severe interstitial disease. Sensitivity of lung ultrasound (100%) was significantly higher than that of chest X-ray (58.33%) for identifying severe interstitial disease (*p* = 0.002) (Table 5).

## 4. Discussion

### 4.1. Summary

Patients in our study most commonly presented fever, followed by cough, in agreement with reports by others [5,10,29]. Both of these symptoms together were significantly associated to positive RT-PCR results, and fever evaluated separately. We observed lung ultrasound had a higher sensitivity than chest X-ray for diagnosing interstitial disease caused by COVID-19 pneumonia in symptomatic patients; this sensitivity increased to 100% for detecting severe interstitial disease. These data highlight the usefulness of lung ultrasound in primary care for early diagnosis of COVID-19-related interstitial disease.

### 4.2. Strengths and Limitations

The retrospective nature of the study may carry an inherent bias. On this note, patients aged 10–18 were excluded here due to disagreements with patients in this age group and their families. However, it may not severely impact conclusions drawn from this study, given that the pediatric population overall was small due to their low COVID-19 infection rate [30]. Additionally, the mean age of patients in this study was 49.62 years, which is similar to that of COVID-19-positive patients reported by others [5]. 

Until mid-May 2020, PCR tests were not available at the primary care centre where this study was carried out; however, patient samples were sent for analysis to the reference hospital, where the same protocol for sample collection was followed. Despite having a relatively small sample size, which still surpassed that of similar studies [23,24,31], the study was able to establish that lung ultrasound was a significantly more accurate method than the chest X-ray.

In any case, further analyses should be considered to confirm obtained results, due to the small sample size of this study.

### 4.3. Comparison with Existing Literature

Several studies have been published on the use of lung ultrasound for the COVID-19 diagnosis; however, most of them were carried out at hospitals and not primary care facilities [23,24,25,26,27,28]. Here, it was highlighted lung ultrasound usefulness in primary care for diagnosing patients in an early phase of COVID-19 infection. A study using a chest X-ray in Spain highlighted its utility in the middle stages of COVID-19 [32], which it is consistent with a chest X-ray having lower sensitivity than the lung ultrasound in an early phase of COVID-19, found in this study. In Lu et al. [24]. It was reported that a high diagnostic efficacy of the lung ultrasound for patients in a severe stage occurred; however, these patients had already been diagnosed with COVID-19, whereas in this study it is shown the utility of lung ultrasound before COVID-19 infection was confirmed.

It was found that the ultrasound had a higher sensitivity than the chest X-ray for early diagnosis of COVID-19 in primary care facilities. This finding is in agreement with a study carried out in an emergency department [25]. Another study compared both methods in a hospital setting but did not evaluate statistical significance [33]. Despite the relatively small sample size used here, this research included a larger patient group than other studies that evaluated the use of lung ultrasound for COVID-19 diagnosis [23,24,31].

### 4.4. Implications for Research and Practice

The RT-PCR test for the COVID-19 diagnosis have several limitations, such as a considerable rate of false negative results [10,11], which are partially due to quality of sampling, the type of sample used, the viral load, and the time after infection at which sample is taken [34]. Despite this, it continues being the best technique to diagnose COVID-19 infection. Currently, we have very useful rapid antigenic tests in primary care for an early diagnosis of COVID-19 infection, however RT-PCR tests and rapid antigenic tests cannot detect if the patient has COVID-19 pneumonia. A quick diagnosis of COVID-19 pneumonia could alleviate the burden on hospitals and accelerate treatment and isolation, when needed.

Although X-rays are a quick and diagnostic method, they present several disadvantages. Carrying out a chest X-rays exposes the patient to radiation [17] and a technician must carry out the X-ray, which may pose time constraints. Identification of interstitial disease in an early phase is challenging with the chest X-ray [19,32], and infected patients may not show as much radiological findings, as evidenced by aforementioned findings of this research.

Lung ultrasound is an easily accessible, accurate, non-invasive and radiation-free diagnostic method [17] that can be useful in the ongoing pandemic. Due to ultrasound’s lack of radiation, it is safe to use and useful for children and pregnant women [31]. Portability of ultrasound machines could be especially useful in containing COVID-19 spread, allowing patients to not leave their home/residence or move inside healthcare centers.

Here, the ultrasound diagnostic tool has greater sensitivity than chest X-ray diagnostic tool to detect COVID-19 pneumonia, however, it has lower specificity. For us, the reason should be found in the greater capacity of the ultrasound tool to detect diseases that cause an interstitial lung syndrome similar to COVID-19 [19]. B lines are also associated with other lung conditions and current knowledge (aside from heart failure) [19] does not allow the physician to distinguish between a “water line B” and a “connective B line” [35]. To improve specificity of the ultrasound technique (avoiding false positives), it should be considered essential to repeat RT-PCR or perform specific serology, and make a correct assessment of symptoms and patient’s history that would orient towards other diseases with a similar ultrasound pattern such as cancer, heart failure, pneumonia, pulmonary fibrosis, tuberculosisTBC and between others [19].

To our knowledge, only another study has investigated a lung ultrasound for early diagnosis of COVID-19 pneumonia in primary care [10]. The study had a smaller sample size than the 212 patients analyzed here and found that a designed severity scale of ultrasound findings was associated with the severity scale of chest X-ray. However, findings were not studied in comparison to the RT-PCR test.

The doctor is very close to the patient when carrying out an ultrasound, which could be a potential risk for doctors examining COVID-19-positive patients. However, to our knowledge, there is currently no literature reporting an event of this nature. To minimize the risk, we carried out a lung ultrasound on the back of patients so that doctors would not be facing them directly. The use of a bedside lung ultrasound has been shown to reduce the number of X-rays and also reduce costs [36]. The use of a lung ultrasound also reduced the number of conventional radiologic exams carried out on patients during the COVID-19 pandemic, which limited exposure of healthcare workers to possible contamination [37].

Although carrying out an ultrasound and interpreting requires training, interstitial lung disease can be identified with precision, given its clear definition by Volpicelli et al. as the presence of at least three B lines in a longitudinal plane between two ribs [19]. On this note, a systematic review found that ultrasound scans carried out by GPs had higher diagnostic accuracy and required less training when examinations were focused within selected anatomic areas [38].

In the ever-changing landscape of the current COVID-19 pandemic, it is critical to be able to make quick decisions in early phases of the disease. Here, it was proven the usefulness of a lung ultrasound in primary care for diagnosing COVID-19 pneumonia in an early phase. Establishing a diagnosis quickly would allow GPs to initiate antibiotic and anticoagulant treatment earlier to protect against a bacterial infection and pulmonary embolism. We believe lung ultrasound is a powerful tool and we hope our findings encourage its adoption by GPs for diagnosing COVID-19 pneumonia.

On the other hand, it was assumed that the greater sensitivity and NPV of pulmonary ultrasound (100%) in the detection of patients with severe interstitial disease should lead primary care teams to carry out more studies searching ultrasound findings, which may be considered as risk factors or protective factors against hospitalization and therefore helping decision-making regarding patients.

## 5. Conclusions

Early diagnosis of COVID-19 pneumonia is of the utmost importance to start treating patients in early stages of the disease. By performing this type of diagnosis, the likelihood of progression to severe states is minimized and patients receive an appropriate treatment.

Taking into consideration that currently the biggest amount of COVID-19 patients is absorbed by primary medical care, where ultrasound makes up a more convenient device to handle, it is imperative that their results turn out relevant, significant and applicable in the daily clinical routine of family doctors.

Consequently this research holds a high level of applicability, since results prove that a lung ultrasound is a more powerful tool compared to chest X-rays to correctly identify patients with COVID-19 pneumonia, especially for those presenting diffuse interstitial disease.

The incorporation of ultrasound as a protocol test in early clinical evaluation of patients with suspected COVID-19 pneumonia constitutes an opportunity to expand the number of cases studied, which will surely help to refine calculations of sensitivity and specificity.

## Figures and Tables

**Figure 1 ijerph-18-03481-f001:**
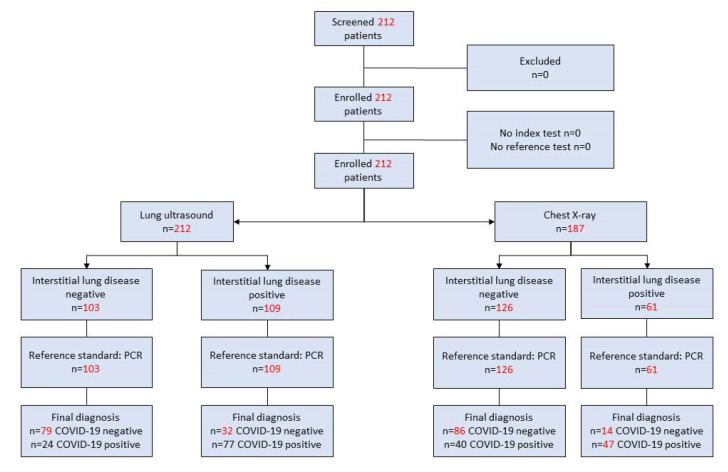
Flow chart documenting participants included in the study.

**Table 1 ijerph-18-03481-t001:** Patient demographic and clinical variables.

Characteristics	*n* (%) *n* = 212
**Age, years (mean ± SD)**	49.62 ± 21.11
**Gender, male**	124 (58.49)
**Symptoms**	
Cough	85 (40.09)
Cough only	27 (12.74)
Fever	153 (72.17)
Fever only	80 (37.74)
Dyspnoea	62 (29.25)
Dyspnoea only	13 (6.13)
Oxygen saturation < 95%	37 (17.45)
Oxygen saturation < 95% only	1 (0.47)
**Hospitalization**	
Yes	25 (11.79)
No	187 (88.21)
**PCR result**	
Positive	101 (47.64)
Negative	111 (52.36)

**Table 2 ijerph-18-03481-t002:** Findings from the lung ultrasound and PCR test.

Ultrasound Pattern	PCR Results *n* (%)	
	RT-PCR+	RT-PCR−
Pathological ultrasound with diffuse interstitial pattern	54 (90.00)	6 (10.00)
Pathological ultrasound with local interstitial pattern	23 (46.92)	26 (53.06)
Normal ultrasound	24 (23.30)	79 (76.70)

**Table 3 ijerph-18-03481-t003:** Relationship between diagnostic findings and COVID-19 infection confirmed by PCR based on hospitalization.

	Normal Ultrasound and Normal Chest X-ray *n* (%)		Normal Ultrasound and Abnormal Chest X-ray *n* (%)		Abnormal Ultrasound and Normal Chest X-ray *n* (%)		Abnormal Ultrasound and Abnormal Chest X-ray *n* (%)	
	RT-PCR+	RT-PCR−	RT-PCR+	RT-PCR−	RT-PCR+	RT-PCR−	RT-PCR+	RT-PCR−
Hospitalized patients (*n* = 24)	0	0	0	0	10 (19.61)	0	14 (28.00)	0
Not hospitalized patients (*n* = 163)	11 (14.67)	64 (85.33)	4 (36.36)	7 (63.64)	19 (37.25)	22 (43.14)	29 (58.00)	7 (14.00)

**Table 4 ijerph-18-03481-t004:** Comparative study of the lung ultrasound and chest X-ray in early detection of COVID-19-related interstitial disease.

	Diagnostic Method (*n* = 187) Value (95% CI)		Statistical Analysis	
	Lung Ultrasound	Chest X-ray	Fisher	McNemar
**Sensitivity**	82.75% (72.83–89.71%)	54.02% (43.04–64.64%)	*p* < 0.001	*p* < 0.001 (4.177)
**Specificity**	71.00% (60.93–79.42%)	86.00% (77.28–91.85%)	*p* = 0.008	*p* = 0.009 (2.599)
**PPV**	71.28% (61.29–79.63%)	77.04% (64.19–86.45%)	-	-
**NPV**	82.55% (72.54–89.59%)	68.25% (59.27–76.09%)	-	-

Abbreviations: 95% CI, 95% confidence interval.

**Table 5 ijerph-18-03481-t005:** Comparative study of lung ultrasound and chest X-ray in early detection of COVID-19-related severe interstitial disease.

	Diagnostic Method (*n* = 187) Value (95% CI)		Statistical Analysis	
	Lung Ultrasound	Chest X-ray	Fisher	McNemar
**Sensitivity**	100% (82.82–100%)	58.33% (36.94–77.20%)	*p* = 0.002	-
**Specificity**	52.76% (44.81–60.56%)	71.16% (63.46–77.84%)	*p* < 0.001	*p* < 0.001 (4.021)
**PPV**	23.76% (16.09–33.45%)	22.95% (13.54–35.80%)	-	-
**NPV**	100% (94.67–100%)	92.06% (85.52–95.91%)	-	-

## Data Availability

This study did not report any data.

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
