# Peer review of "Higher Accuracy of Lung Ultrasound over Chest X-ray for Early Diagnosis of COVID-19 Pneumonia"

_ijerph, 2021, doi:10.3390/ijerph18073481_

Round 1
Reviewer 1 Report
The research study is very interesting because of the actuality of
the corona pandemic. It promotes alternative diagnostic elements to
expedite the crisis management process. The chapter "Statistical Analysis" is not very clear at all major stages. So it would be great, ifthe author could improve this in the article. Another point is the
chapter "Strength and Limits". It would be great to point out more
that this study only focuses on a small section. It would also be
desirable to show perspectives on how this survey can be raised to a broader level. This would ensure that the knowledge can be used.
Furthermore, it would then also show whether the results obtained
here are also visible in a broader study.
Author Response
Reviewer 1
Question (Q):
The research study is very interesting because of the actuality of the corona pandemic. It promotes alternative diagnostic elements to expedite the crisis management process.
The chapter "Statistical Analysis" is not very clear at all major stages. So it would be great, if the author could improve this in the article.
Answer (A):
It has been changed in the new version of the manuscript. In page 3, line 143-153.
Q
Another point is the chapter "Strength and Limits". It would be great to point out more
that this study only focuses on a small section.
A
It has been changed in the new version of the manuscript. In page 8, line 244-254.
Q
It would also be desirable to show perspectives on how this survey can be raised to a broader level. This would ensure that the knowledge can be used.
A
It has been changed in the new version of the manuscript. In page 9, line 273-313.
Q
Furthermore, it would then also show whether the results obtained here are also visible in a broader study.
A
We have included in the new version of the manuscript. In page 9, line 314.
Reviewer 2 Report
Manuscript ID: ijerph-1135012
Title: Higher accuracy of lung ultrasound over chest X-ray for early COVID-19 diagnosis
Authored by Martinez Redondo and colleagues (corresponding author Palacin Peruga)
General comments:
The paper is generally well-written, to the point. The tables with the larger-than needed line spacing can be distracting. Since spacing (real estate) in a journal is often at a premium, using single-line spacing may reduce the entire manuscript by one page. There are a few issues that needs to be addressed.
Issues to address:
The discussion should have a section to address issues related with the high false-positive rate of the method caused by its better sensitivity but lower specificity. Authors should address the pros and cons of this issue and whether they consider alternatives or modifications to improve specificity. B lines are also associated with other lung conditions, particularly edema/fluid cumulation, not just the diagnosis of diffuse interstitial patterns, mostly referring to fibrosis/scarring of the pulmonary tissues. Current knowledge does not allow the physician to distinguish between a “water B-line” and a “connective B-line” (Soldati and Demi, 2017).
Line 288 is overreaching. The authors cited many ultrasound studies related to COVID, and several were left out. Several studies addressed the values of ultrasounds for diagnosis. Further, quick PCR are getting better, and results might be obtained before an ultrasound is performed and diagnosis reached, particularly in settings were technicians, not physicians, are performing the ultrasounds. Pushing the value of early diagnosis using ultrasound is kind of fishy particularly with the low specificity and the presence of false negative also. The authors should modify that section of the discussion to highlight the value of the technique for early screening before a definitive diagnosis, since there is likely a good correlation between the presence of B lines and severity of the disease and the likelihood of hospitalization and adverse outcomes. The technique could be used for early triage of symptomatic patients.
Minor issues:
Line 289: Should say “The study had” instead of “This” since it is referring to the paper cited above and not the current manuscript.
Once defined, abbreviations (PCR, PPV, NPV, etc.) do not need to be spelled-out again, including below tables. Even PCR is often not even defined in many scientific journals, having reached the same standing as DNA or other abbreviations not needing definition.
Author Response
Reviewer 2
Question (Q):
General comments:
The paper is generally well-written, to the point. The tables with the larger-than needed line spacing can be distracting. Since spacing (real estate) in a journal is often at a premium, using single-line spacing may reduce the entire manuscript by one page. There are a few issues that needs to be addressed.
Answer (A):
It has been changed in the new version of the manuscript. In pages 4-8.
Q
Issues to address:
The discussion should have a section to address issues related with the high false-positive rate of the method caused by its better sensitivity but lower specificity. Authors should address the pros and cons of this issue and whether they consider alternatives or modifications to improve specificity. B lines are also associated with other lung conditions, particularly edema/fluid cumulation, not just the diagnosis of diffuse interstitial patterns, mostly referring to fibrosis/scarring of the pulmonary tissues. Current knowledge does not allow the physician to distinguish between a “water B-line” and a “connective B-line” (Soldati and Demi, 2017).
A
We have included the reference in the new version of the manuscript. In page 9, lines 299-311.
Q
Line 288 is overreaching. The authors cited many ultrasound studies related to COVID, and several were left out. Several studies addressed the values of ultrasounds for diagnosis. Further, quick PCR are getting better, and results might be obtained before an ultrasound is performed and diagnosis reached, particularly in settings were technicians, not physicians, are performing the ultrasounds. Pushing the value of early diagnosis using ultrasound is kind of fishy particularly with the low specificity and the presence of false negative also. The authors should modify that section of the discussion to highlight the value of the technique for early screening before a definitive diagnosis, since there is likely a good correlation between the presence of B lines and severity of the disease and the likelihood of hospitalization and adverse outcomes. The technique could be used for early triage of symptomatic patients.
A
Despite making clear the aim of this study, is absolutely right in his comments. Probably this point is not clear enough. We do not intend to replace diagnostic tests for COVID-19 infection (RT-PCR or Antigenic Test) by lung ultrasound. Our objective is to compare ultrasound vs radiology in the detection of COVID-19 pneumonia. RT-PCR and antigenic tests can diagnose COVID-19 infection, but not the presence of lung disease in patients.
It has been changed in the new version of the manuscript.
Minor issues:
Q
Line 289: Should say “The study had” instead of “This” since it is referring to the paper cited above and not the current manuscript.
A
It has been changed in the new version of the manuscript.
Q
Once defined, abbreviations (PCR, PPV, NPV, etc.) do not need to be spelled-out again, including below tables. Even PCR is often not even defined in many scientific journals, having reached the same standing as DNA or other abbreviations not needing definition.
A
It has been changed in the new version of the manuscript.